

# Exogenous melatonin improves salt stress adaptation of cotton seedlings by regulating active oxygen metabolism

Dan Jiang[1,2,*], Bin Lu[3,*], Liantao Liu[2], Wenjing Duan[1,2], Li Chen[1,2], Jin Li[1], Ke Zhang[2], Hongchun Sun[2], Yongjiang Zhang[2], Hezhong Dong[2,4], Cundong Li[2] and Zhiying Bai[1,2]

[1] State Key Laboratory of North China Crop Improvement and Regulation/College of Life Science, Hebei Agricultural University, Baoding, China
[2] State Key Laboratory of North China Crop Improvement and Regulation/Key Laboratory of Crop Growth Regulation of Hebei Province/College of Agronomy, Hebei Agricultural University, Baoding, China
[3] College of Landscape and Tourism, Hebei Agricultrual University, Baoding, China
[4] Cotton Research Center/Key Laboratory of Cotton Breeding and Cultivation in Huang-huai-hai Plain, Ministry of Agriculture, Shandong Academy of Agricultural Sciences, Jinan, China
[*] These authors contributed equally to this work.

Corresponding authors
Cundong Li, nxylcd@hebau.edu.cn, auhlcd@163.com
Zhiying Bai, zhiyingbai@126.com

## ABSTRACT

Melatonin is a small-molecule indole hormone that plays an important role in participating in biotic and abiotic stress resistance. Melatonin has been confirmed to promote the normal development of plants under adversity stress by mediating physiological regulation mechanisms. However, the mechanisms by which exogenous melatonin mediates salt tolerance via regulation of antioxidant activity and osmosis in cotton seedlings remain largely unknown. In this study, the regulatory effects of melatonin on reactive oxygen species (ROS), the antioxidant system, and osmotic modulators of cotton seedlings were determined under 0–500 $\mu$M melatonin treatments with salt stress induced by 150 mM NaCl treatment. Cotton seedlings under salt stress exhibited an inhibition of growth, excessive hydrogen peroxide ($H_2O_2$), superoxide anion ($O_2^-$), and malondialdehyde (MDA) accumulations in leaves, increased activity levels of superoxide dismutase (SOD), peroxidase (POD), catalase (CAT), and ascorbate peroxidase (APX), and elevated ascorbic acid (AsA) and glutathione (GSH) content in leaves. However, the content of osmotic regulators (i.e., soluble sugars and proteins) in leaves was reduced under salt stress. This indicates high levels of ROS were produced, and the cell membrane was damaged. Additionally, osmotic regulatory substance content was reduced, resulting in osmotic stress, which seriously affected cotton seedling growth under salt stress. However, exogenous melatonin at different concentrations reduced the contents of $H_2O_2$, $O_2^-$, and MDA in cotton leaves, increased the activity of antioxidant enzymes and the content of reductive substances (i.e., AsA and GSH), and promoted the accumulation of osmotic regulatory substances in leaves under salt stress. These results suggest that melatonin can inhibit ROS production in cotton seedlings, improve the activity of the antioxidant enzyme system, raise the content of osmotic regulation substances, reduce the level of membrane lipid peroxidation, and protect the integrity of the lipid membrane under salt stress, which reduces damage caused by salt stress to seedlings and effectively enhances inhibition of salt stress on cotton seedling growth. These results indicate that 200 $\mu$M melatonin treatment has the best effect on the growth and salt tolerance of cotton seedlings.

## INTRODUCTION

As the ecological environment continues to change at a global scale, the area of the world's saline-alkali land has increased annually, and soil salinization has thus become a major impediment to agricultural development (*Munns, 2002*). Salt stress brings ionic stress and secondary stresses such as osmotic stress and oxidative stress to plants, which interfere with the normal development of plants. When plants are subjected to salt stress, their physiological and biochemical reactions cannot proceed normally, leading to reduced yields and quality and even plant death (*Parida & Das, 2005*; *Liang et al., 2018*). Cotton is the field crop with the longest planting industry chain in the world, and it plays an important role in national economies (*Luo et al., 2018*). The seedling stage is a fragile and critical stage in the growth and development of cotton, which is severely affected by salt stress, so the exploration of the growth and regulation of cotton seedlings under salt stress is of great importance.

Melatonin (*N*-acetyl-5-methoxytryptamine, MT) plays an important role in promoting plant development and responses to abiotic stress (*Ren, Rutto & Katuuramu, 2019*). Melatonin was first isolated from the pineal gland of cattle and was subsequently discovered in vascular plants in 1995 (*Dubbels et al., 1995*; *Hattori et al., 1995*). The *N*-acetyl and 5-methoxy functional groups in the structure of melatonin not only establish its high lipophilicity and hydrophilicity, but also determine the specificity of its binding to receptors (*Arnao & Hernández-Ruiz, 2015*). Isotopic tracing experiments have confirmed that melatonin is synthesized in plants (*Murch, Krishnaraj & Saxena, 2000*), and it is believed that mitochondria and chloroplasts in plants are the synthesis sites of melatonin, and in most organisms, melatonin can be transferred from mitochondria and chloroplasts to other tissues and organs (*Tan et al., 2013*). Subsequent studies have shown that melatonin is widely present in almost all plant species, and people have also found melatonin in various organs of higher plants (*Kolar & Machackova, 2005*; *Okazaki & Ezura, 2009*; *Saeteaw et al., 2013*), at concentrations ranging from 0.1 pg g$^{-1}$ (FW) to 20~30 µg g$^{-1}$ (FW) (*Hardeland, 2016*) and even 230 µg g$^{-1}$ (DW) (*Oladi et al., 2014*). The biosynthetic precursor of melatonin is tryptophan, which has a function similar to that of auxin (*Arnao & Hernández-Ruiz, 2018*). Subsequent studies have shown that melatonin plays an important role in the regulation of seed germination (*Tiryaki & Keles, 2012*), plant growth (*Manchester et al., 2015*), cell division (*Park et al., 2012*), delaying leaf senescence (*Byeon et al., 2012*; *Wang et al., 2013*), and defense against abiotic stresses such as extreme temperature (*Ahammed et al, 2019*), heavy metals (*Kaya et al., 2019*; *Posmyk et al., 2008*), UV radiation (*Afreen, Zobayed & Kozai, 2006*), salinity (*Zhang et al., 2015*), and drought (*Sharma & Zheng, 2019*).

As an endogenous free radical scavenger, melatonin not only directly neutralizes reactive oxygen species (ROS) and reactive nitrogen species (RNS), but also stimulates antioxidant enzymes, thereby improving its antioxidant efficiency (*Pieri et al., 1994*; *Reiter et al., 2000*).

Melatonin has also been shown to scavenge ROS efficiently in vivo using transgenic plants, leading to oxidative stress resistance (*Park et al., 2013*). Exogenous melatonin increases the activity of superoxide dismutase (SOD; EC 1.15.1.1), peroxidase (POD; EC 1.11.1.7), catalase (CAT; EC 1.11.1.6), and ascorbate peroxidase (APX; EC 1.11.1.11), thus enhancing the antioxidant capacity and reducing the effects of cold and water stress on tea trees (*Li et al., 2018a*) and cucumber seedlings (*Zhang et al., 2013*). Ascorbic acid (AsA) and glutathione (GSH) are important reducing substances in plants, where they provide defense against membrane lipid peroxidation. The AsA-GSH cycle system can effectively scavenge free radicals; in apples, melatonin treatment can maintain higher AsA and GSH contents, reduce dehydroascorbic acid (DHA) and oxidized glutathione (GSSG) content, and delay the senescence of leaves in dark conditions (*Wang et al., 2012*). Plants mainly resist salt stress damage through osmotic regulation (*Hu et al., 2018*). Studies have shown that soluble sugar and soluble protein are important osmotic regulation substances in increasing the concentration of cell fluid and improving the resistance of plants to salt stress (*Park, Kim & Yun, 2016*). Melatonin treatment reduces cellular damage by increasing the contents of proline, soluble proteins, and soluble sugars in melon, indicating that exogenous melatonin can enhance the low-temperature adaptability of melon by increasing the content of osmotic substances (*Gao et al., 2016*).

We have previously studied the regulatory effect of exogenous melatonin on cotton seed germination (*Xiao et al., 2019*; *Chen et al., 2020*), but the regulation and control of exogenous melatonin on cotton seedlings under salt stress remains unknown. Therefore, in this study, seeds of the cotton cultivar 'Guoxin 9' were employed (1) to examine the effects of different concentrations of exogenous melatonin on osmotic substance content and oxidoreductase activity under salt stress, (2) to explore the mechanism by which exogenous melatonin mitigates impairment of cotton seedling growth under salt stress, and (3) to screen for the optimal melatonin concentration for ameliorating salt stress. This study aims to provide a theoretical basis for the development and utilization of melatonin and the cultivation of salt-resistant cotton.

## MATERIALS & METHODS

### Reagents
Melatonin (*N*-acetyl-5-methoxytryptamine) was obtained from Sigma-Aldrich (St. Louis, MO, USA). All other reagents used in all experiments were of analytical grade.

### Plant material
A conventional, widely planted transgenic insect-resistant cotton (*Gossypium hirsutum* L.) cultivar, 'Guoxin No.9,' was used in this study, which was provided by Guoxin Rural Technical Service Association. The experiment was conducted in the greenhouse facilities of Hebei Agricultural University, Baoding City, Hebei Province, China (38.85°N, 115.30°E) from March 2019 to April 2020.

### Experimental design
Cotton (*Gossypium hirsutum* L.) seeds were sterilized with 0.1% $HgCl_2$ (*w/v*) for 10 min, followed by three washes with sterile distilled water, to remove any residual disinfectant, and

then germinated in an incubator at 25 °C for 24 h. The germinated seeds were sown in plugs of vermiculite and cultivated in the greenhouse. The seedlings were cultivated at 28/25°C (day/night) with a relative humidity of $45 \pm 5\%$, and at a photoperiod (600 $\mu$molm$^{-2}$ s$^{-1}$ light intensity) of 16 h/8 h (day/night). At the two-true-leaf stage, the seedlings were transferred to hydroponic tubes containing 1/4 strength complete nutrient solution for hydroponic culture (in a PVC drum, bottom diameter 110 cm, height 200 cm), which was then replaced with half-strength complete nutrient solution for 4 d. Afterwards, seedlings were transferred to full-strength nutrient solution for continued cultivation until the end of the experiment (*Wang et al., 2020*). The complete nutrient solution for cotton was modified Hoagland solution (*Mengel, Robin & Salsac, 1983*), consisting of 5 mM KNO$_3$, 2 mM MgSO$_4$·7H$_2$O, 1 mM KH$_2$PO$_4$, 4.5 mM NH$_4$H$_2$PO$_4$, 5 mM Ca(NO$_3$)$_4$·4H$_2$O, 0.1 mM EDTA-Na$_2$, 0.1 mM FeSO$_4$·7H$_2$O, and micronutrients (5 $\mu$M H$_3$BO$_3$, 7 $\mu$M MnSO$_4$ ·H$_2$O, 0.8 $\mu$M ZnSO$_4$ ·7H$_2$O, 0.3 $\mu$M CuSO$_4$ ·5H$_2$O, and 0.02 $\mu$M (NH$_4$)$_6$Mo$_7$O$_{24}$·4H$_2$O). Cotton seedlings were fixed into holes with a sponge when transplanted and aerated with a small air pump for 1 h each day. When the cotton seedlings reached the three-true-leaf stage 6 d after transplantation, the following treatments were imposed.

Seedlings were treated with exogenous MT at various concentrations and subjected to a salt stress treatment; specifically, the cotton seedlings were sprayed respectively with 50, 100, 200, and 500 $\mu$M MT solutions, until the leaves dripped, once every 24 h for 12d. The following experimental groups were arranged into a randomized complete block design with 30 replicates and treated as follows: (1) no melatonin/no salt treatment (control, CK); (2) no melatonin/150 mM NaCl treatment (150 mM NaCl, as determined by the pretest screening) (S); (3) 50 $\mu$M MT-applied/150 mM NaCl treatment (S+MT50); (4) 100 $\mu$M MT-applied/150 mM NaCl treatment (S+MT100); (5) 200 $\mu$M MT-applied/150 mM NaCl treatment (S+MT200); and (6) 500 $\mu$M MT-applied/150 mM NaCl treatment (S+MT500). Nutrient solutions for each treatment were renewed every 2 d, and 30 PVC hydroponic drums were used for each treatment. The physiological indexes of cotton plants were measured using the third functional leaf (from the top of the plant) at 0, 3, 6, 9, and 12 d after treatment.

## Determination of plant height and leaf area

After treatment, six cotton seedlings were randomly selected at 3, 6, 9, and 12 d for plant height measurements. At the same time, the green leaf area of the whole plant was calculated using the length and width coefficient method (0.75) based on measurements with a ruler.

## Determination of ROS and MDA content

Hydrogen peroxide (H$_2$O$_2$) content was determined by using a H$_2$O$_2$ assay kit (A064, Nanjing Jiancheng Bioengineering Institute, Nanjing, China). Superoxide anion (O$_2^-$) production rate was determined by using a O$_2^-$ assay kit (SA-1-G, Suzhou Comin Biotechnology Co., Ltd., Suzhou, China). To detect lipid peroxidation and membrane integrity, the malondialdehyde (MDA) content was measured by using a MDA assay kit (MDA-1-Y, Suzhou Comin Biotechnology Co., Ltd., Suzhou, China) according to the manufacturer's instructions.

### Determination of antioxidant enzyme activity

Superoxide dismutase (SOD), peroxidase (POD), catalase (CAT), and ascorbate peroxidase (APX) activity levels were determined by using assay kits for each respective enzyme (SOD-1-Y, POD-1-Y, CAT-1-W, and APX-1-W, respectively, Suzhou Comin Biotechnology Co., Ltd. Suzhou, China) according to the manufacturer's instructions.

### Determination of antioxidant metabolites

AsA and GSH contents were measured by using AsA and GSH assay kits, respectively (AsA-1-W and GSH-1-W, Suzhou Comin Biotechnology Co., Ltd.) according to the manufacturer's instructions.

### Determination of soluble sugar and protein, two osmotic regulators

Soluble sugar content and soluble protein content were determined by using a soluble sugar assay kit (A145-1-1, Nanjing Jiancheng Bioengineering Institute) and a BCA assay kit (BCAP-1-W, Suzhou Comin Biotechnology Co., Ltd.) according to the manufacturer's instructions.

### Statistical analysis

All experimental data were performed using IBM SPSS Statistics 21.0 software and reported as mean $\pm$ standard deviation (SD) values. A statistical significance threshold of $P < 0.05$ was used to evaluate one-way ANOVA results.

## RESULTS

### Exogenous melatonin affects the growth of cotton seedlings under salt stress

As can be seen from Fig. 1A, salt stress (150 mM NaCl) inhibited the growth of cotton seedlings compared with control plants (CK). Compared with salt stress (S) alone, the treatment with 50, 100, 200, and 500 μM exogenous melatonin reduce the inhibition of cotton seedlings growth under salt stress. Figures 1B–1C shows the trends in plant height and leaf area were quite similar, increasing gradually over time. Under salt stress (S), plant height was significantly reduced compared with that of the control (CK), with decreases of 16.2%, 23.5%, 22.5%, and 25% by 3, 6, 9, and 12 d, respectively (Fig. 1B). After the application of melatonin, the height of plants increased first and then decreased with the increase of melatonin concentration. Under 200 μM melatonin treatments, the heights of cotton plants across the different periods (3 d, 6 d, 9 d, and 12 d) were highest, 9.87 cm, 11.05 cm, 12.2 cm, and 14.47 cm, respectively, and compared with S plants, they were significantly increased by 19.4%, 21.2%, 19.4%, and 25.4%, respectively. However, there was no significant difference, indicating that 200 μM melatonin effectively promoted the increase of cotton plant height.

Under salt stress, the leaf area was significantly reduced, by 24%, 28.7%, 25.9%, and 24.9%, compared with CK at 3, 6, 9, and 12 d, respectively (Fig. 1C). When treated with different concentrations of melatonin, the leaf area in cotton leaves showed different trends. When treated with 50 μM melatonin at 9, and 12 d, the leaf area of cotton seedlings was significantly higher than that of S plants. When the melatonin concentration was 200 μM,

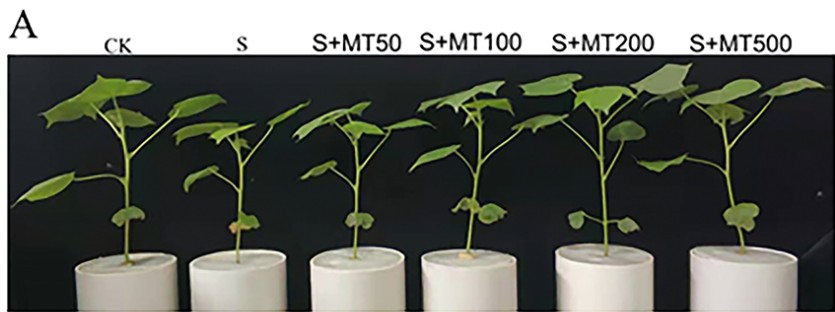

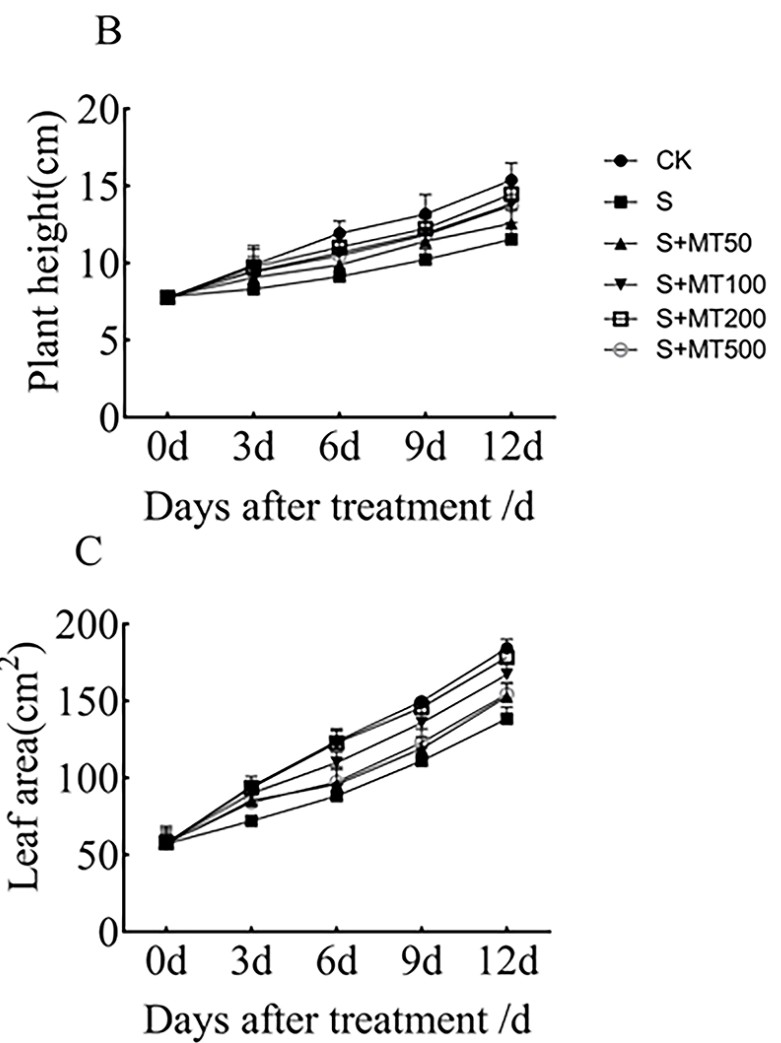

**Figure 1** **Effects of exogenous melatonin (MT) on cotton seedling phenotype (A), plant height (B), and leaf area (C) of cotton seedlings.** Control (CK) and salt-treated (S) plants were sprayed with distilled water, while the S + MT50, S + MT100, S + MT200, and S + MT500 plants were sprayed with 50, 100, 200, 500 μM MT, respectively.

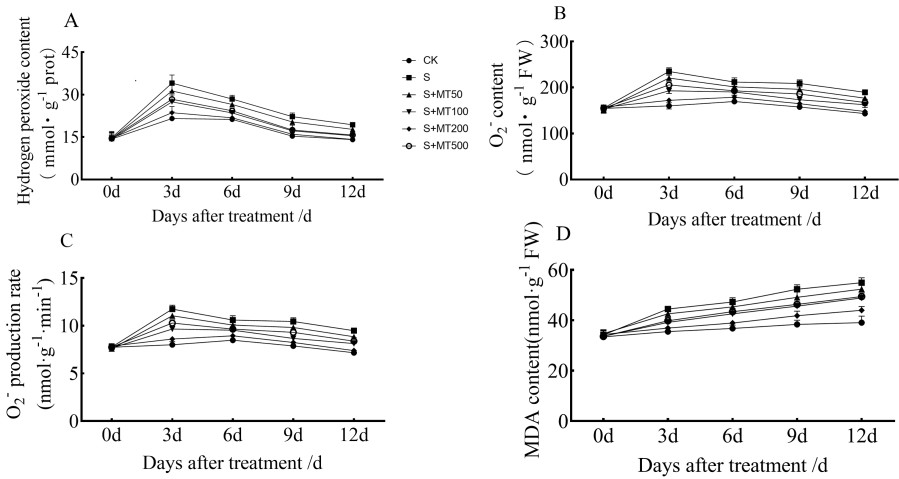

**Figure 2  Effects of exogenous melatonin (MT) on hydrogen peroxide ($H_2O_2$) (A), superoxide anion ($O_2^-$) (B), superoxide anion production rate (C), and malondialdehyde (MDA) (D) content in cotton leaves under salt stres.** Control (CK) and salt-treated (S) plants were sprayed with distilled water, while the S + MT50, S + MT100, S + MT200, and S + MT500 plants were sprayed with 50, 100, 200, 500 $\mu$M MT, respectively.

the increase was greatest. At this concentration, the leaf areas on 3 d, 6 d, 9 d, and 12 d were 93.65 cm², 121.16 cm², 145.69 cm², and 178.32 cm², respectively; compared with S plants, they were significantly increased, by 29.8%, 37.2%, 31.2%, and 28.8%, respectively, indicating that 200 $\mu$M melatonin effectively promoted the increase of leaf area.

## Exogenous melatonin affects ROS content of cotton seedlings under salt stress

When plants are under stress, high levels of reactive oxygen species ($H_2O_2$, $O_2^-$, etc.) are accumulated, causing oxidative stress. MDA, a metabolite of membrane lipid peroxidation, can reflect the degree of cellular damage (*Farooq et al., 2017*).

Figures 2A–2C shows that the trends in contents of $H_2O_2$, $O_2^-$, and superoxide anion production rates of leaves were quite similar, and decreased gradually over time. However, the MDA content increased gradually over time, reaching their highest values at 12 d (Fig. 2D).

Compared with the CK treatment, the $H_2O_2$ content under the salt stress (S) treatment increased significantly, by 58.1%, 33.9%, 45.1%, and 37.2% at 3, 6, 9, and 12 d, respectively (Fig. 2A). When treated with melatonin treatments, the $H_2O_2$ content was significantly lower than that of S plants. With exogenous melatonin, $H_2O_2$ content first decreased and then increased as melatonin concentration increased. Under the 50 $\mu$M melatonin treatment, the $H_2O_2$ content of cotton leaves was lower than that of S plants during the treatment period, but there was no significant difference; under the 100, 200, and 500 $\mu$M melatonin treatments at 3, 6, 9, and 12 d, the $H_2O_2$ content was significantly lower than that of S plants. Among them, the $H_2O_2$ content of cotton leaves decreased most significantly under 200 $\mu$M melatonin treatment, and the $H_2O_2$ content at 3, 6, 9, and 12

d was decreased by 30.7%, 23.4%, 28%, and 26.8%, respectively, indicating that 200 $\mu$M melatonin had the most obvious effect on reducing the $H_2O_2$ content of cotton leaves.

Compared with the CK treatment, the $O_2^-$ content under the salt stress (S) treatment increased significantly, by 46.8%, 24.9%, 32.6%, and 32.3% at 3, 6, 9, and 12 d, respectively (Fig. 2B). When exogenous melatonin was applied, the $O_2^-$ content first decreased and then increased as melatonin concentration increased. Under the 50 $\mu$M melatonin treatment, the $O_2^-$ content of cotton leaves was significantly higher than that of S plants at 3, 9, and 12 d. Under 100, 200 and 500 $\mu$M melatonin treatments, $O_2^-$ content was significantly higher than that of S plants at 3, 6, 9, and 12 d, with the 200 $\mu$M melatonin treatment inducing the most significant decrease in $O_2^-$ content. $O_2^-$ content levels were reduced by 26.8%, 15.6%, 21%, and 22%, respectively, at 3, 6, 9, and 12 d, compared with S plants, indicating that the 200 $\mu$M melatonin treatment had the most inhibitory effect on $O_2^-$ accumulation of cotton leaves. In addition, the trend in the superoxide anion production rate of leaves is similar to that of $O_2^-$ content, and the results also showed that 200 $\mu$M melatonin treatment effectively inhibited superoxide anion production in cotton leaves (Fig. 2C).

Compared with CK plants, the MDA content of cotton seedlings under salt stress (S) increased significantly, by 25.2%, 28.6%, 36.4%, and 40.5% at 3, 6, 9, and 12 d, respectively (Fig. 2D). Under exogenous melatonin, MDA first decreased and then increased as melatonin concentration increased. Under the 50 $\mu$M melatonin treatment, the MDA content of cotton seedlings was significantly lower than that of S plants only at 3, 6, and 9 d; under the 100, 200, and 500 $\mu$M melatonin treatments, the MDA content was significantly lower than that of S plants at 3, 6, 9, and 12 d, with MDA decreased most significantly under the 200 $\mu$M melatonin treatment. The MDA contents at 3, 6, 9, and 12 d were decreased, by 17%, 17.7%, 20.1%, and 20%, compared with S plants, indicating that the 200 $\mu$M melatonin treatment significantly inhibited the accumulation of MDA content in cotton leaves.

## Exogenous melatonin affects antioxidant enzymes of cotton seedlings under salt stress

Under adverse conditions, plants use their own enzymatic antioxidant system (i.e., SOD, POD, CAT, and APX) to remove excess ROS so as to protect cells from oxidative damage caused by these conditions (*Munns & Tester, 2008*).

Figure 3 shows the trends in SOD, POD, CAT, and APX activity levels of leaves under control (CK) and salt stress (S) conditions are quite similar, with no significant change over time under the control (CK) treatment. However, the SOD, POD, and CAT activity levels of leaves under salt stress (S) decreased over time, reaching their lowest values at 12 d.

The SOD activity under the salt stress (S) treatment was significantly increased, with SOD activities at 3, 6, 9, and 12 d that were 46.6%, 37.3%, 26.1%, and 18.2% higher than those of CK plants, respectively (Fig. 3A). After applying different concentrations of melatonin, the SOD activity in cotton leaves increased first and then decreased. When treated with 50, 100, and 500 $\mu$M melatonin at 3, 6, 9, and 12 d, the SOD activity of cotton

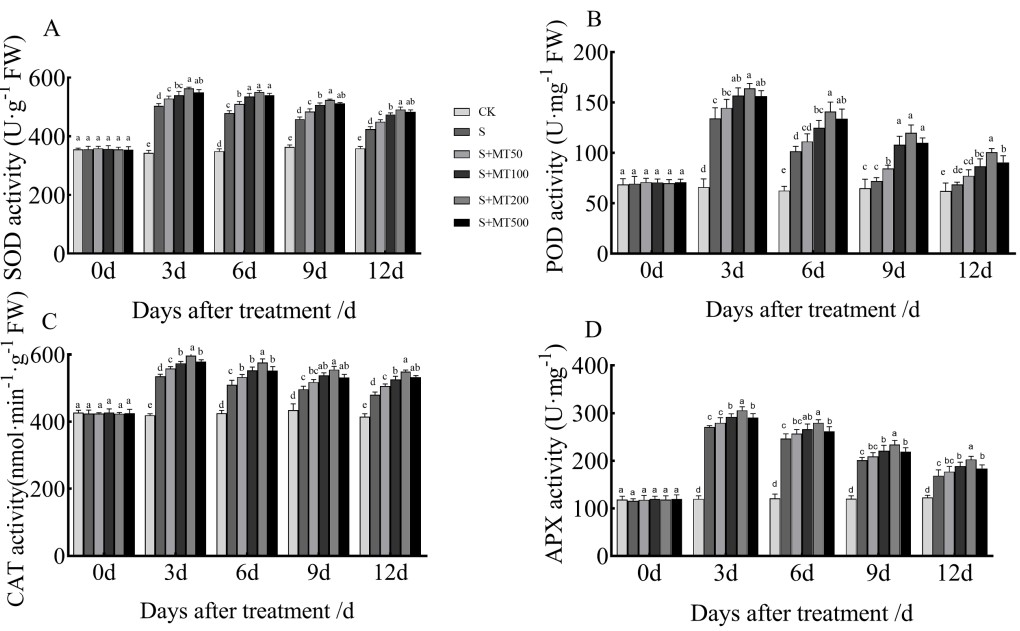

**Figure 3  Effects of exogenous melatonin (MT) treatment on superoxide dismutase (SOD) (A), peroxidase (POD) (B), catalase (CAT) (C), and ascorbate peroxidase (APX) (D) activities of cotton leaves under salt stress.** Control (CK) and salt-treated (S) plants were sprayed with distilled water, while S + MT50, S + MT100, S + MT200, and S + MT500 plants were sprayed with 50, 100, 200, and 500 μM MT, respectively. Different lowercase letters indicate significant differences at the $P \leq 0.05$ level.

seedlings was significantly higher than that of S, with the 200 μM melatonin treatment having the most obvious effect in increasing SOD activity. The SOD activities at 3, 6, 9, and 12 d were increased by 11. 9%, 14.9%, 14.2%, and 15.9%, respectively, compared with S plants, indicating that 200 μM melatonin had the most obvious effect promoting the SOD activity of cotton leaves.

The POD activity under the salt stress (S) treatment was significantly increased (Fig. 3B), with POD activity levels at 3, 6, 9, and 12 d that were increased by 103.2%, 62%, 11.2%, and 10.6%, respectively, compared with the CK plants. After applying melatonin at different concentrations, the variation trend of POD was similar to that of SOD. When treated with 50 μM melatonin at 9, and 12 d, the POD activity of cotton seedlings was significantly higher than that of S plants. Under 100, 200, and 500 μM melatonin treatments across 3, 6, 9, and 12 d, the POD activity was significantly higher than that of S plants; the increase in POD activity of cotton seedlings was the most significant under the 200 μM melatonin treatment. When compared with S plants, The POD activities at 3, 6, 9, and 12 d were increased by 22.3%, 39%, 66.3%, and 46.4%, respectively, indicating that 200 μM melatonin most obviously promoted the POD activity of cotton leaves.

The CAT activity under the salt stress (S) treatment was significantly increased, and at 3, 6, 9, and 12 d, it was increased by 27.8%, 19.9%, 14.3%, and 15.7%, respectively, compared with CK plants (Fig. 3C). When treated with 50 μM melatonin at 3, 9, and 12 d, the CAT activity of cotton seedlings was significantly higher than that of S plants. Under 100, 200,

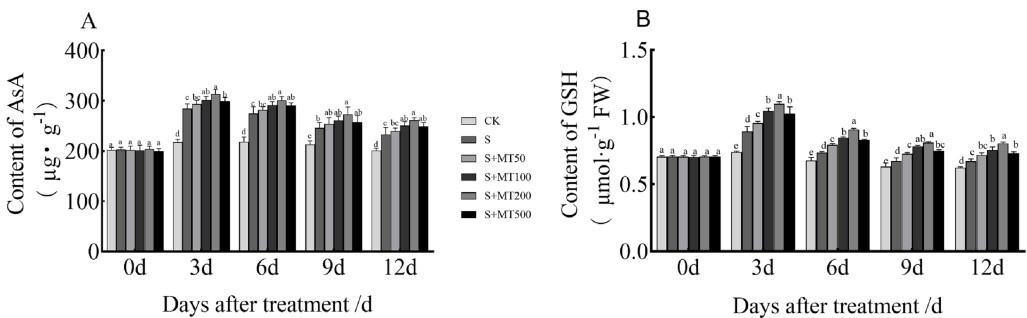

**Figure 4 Effects of exogenous melatonin (MT) treatment on ascorbic acid (AsA) (A) and glutathione (GSH) (B) contents of cotton leaves under salt stress.** Control (CK) and salt-treated (S) plants were sprayed with distilled water, while S + MT50, S + MT100, S + MT200, and S + MT500 plants were sprayed with 50, 100, 200, and 500 μM MT, respectively. Different lowercase letters indicate significant differences at a $P \leq 0.05$ threshold.

and 500 μM melatonin treatments at 3, 6, 9, and 12 d, the CAT activity was significantly higher than that of S plants. Among them, the increase in CAT activity was most obvious under the 200 μM melatonin treatment. The CAT activity levels at 3, 6, 9, and 12 d were 11.5%, 13%, 11.8%, and 14.4% higher, respectively, than those of S plants, indicating that 200 μM melatonin most obviously promoted the CAT activity of cotton leaves.

The APX activity under the salt stress (S) treatment was significantly increased, and at 3, 6, 9, and 12 d, it was increased by 125.9%, 104.2%, 67.2%, and 37.1%, respectively, compared with CK plants (Fig. 3D). After applying melatonin, the APX activity increased. Under the 200 μM melatonin treatment, the APX activity levels at 3, 6, 9, and 12 d were 12.9%, 13. 5%, 16.5%, and 20.2% higher, respectively, than those of S plants, indicating that 200 μM melatonin most obviously promoted the APX activity of cotton leaves.

## Exogenous melatonin affects AsA and GSH contents of cotton seedlings under salt stress

Figure 4 shows the trends in AsA and GSH content levels under control (CK) ere quite uniform, with no significant change over time under control (CK) conditions. However, the AsA and GSH content levels of leaves under salt stress (S) decreased over time, reaching their lowest values at 12 d. The AsA content under the salt stress (S) treatment was significantly increased, and the AsA content at 3, 6, 9, and 12 d increased by 30.8%, 25.2%, 15.7%, and 15. 9%, respectively, compared with CK plants (Fig. 4A). Under the 50 μM melatonin treatment, the AsA content was higher than that of S plants throughout the treatment period, but there was no significant difference. Under the 100 and 500 μM melatonin treatments, the AsA content was significantly higher than that of S plants at 3, 6, and 12 d, with the most obvious increase under the 200 μM melatonin treatment. The AsA contents at 3, 6, 9, and 12 d were increased by 10.2%, 10%, 11.4%, and 12.2% compared with S plants; these differences were significant, indicating that 200 μM melatonin treatment effectively increased the AsA content of cotton leaves.

The GSH content under the salt stress (S) treatment was significantly increased, and the GSH content at 3, 6, 9, and 12 d increased by 20.7%, 8.9%, 6.5%, and 7.9%, respectively,

compared with CK plants (Fig. 4B). Under the 50 µM melatonin treatment, the GSH content was significantly higher than that of S plants at 3, 6, and 9 d; under the 100, 200, and 500 µM melatonin treatments, the GSH content was significantly higher than that of S plants at 3, 6, 9, and 12 d; among these treatments, the increase effect was the most obvious under the 200 µM melatonin treatment. The GSH content levels at 3, 6, 9, and 12 d were increased by 23%, 23.3%, 20.3%, and 19.8% respectively, compared with S plants; these significant differences indicate that 200 µM melatonin treatment effectively increased the GSH content of cotton leaves.

## Exogenous melatonin affects organic osmotic substance content of cotton seedlings under salt stress

Salt stress can cause drought stress to plants. To prevent water loss, plants often accumulate various substances to increase the concentrations of cellular fluids. Soluble sugars and proteins are important osmotic regulators in plants (*Kerepesi & Galiba, 2000*; *Yang & Guo, 2018*).

Figure 5 shows the trends in content of soluble sugar and protein in cotton leaves under control (CK) were quite uniform, increasing gradually over time, reaching their maximum values at 12 d. The soluble sugar content under the salt stress (S) treatment was significantly reduced, with soluble sugar contents of leaves at 3, 6, 9, and 12 d decreased by 58.5%, 52.2%, 27.5%, and 33.3%, respectively, compared with CK plants (Fig. 5A). Under exogenous melatonin, the soluble sugar content first increased and then decreased as melatonin concentration increased. Under the 50 µM melatonin treatment, the soluble sugar content of cotton seedlings was significantly higher than that of S plants only at 3 d; under the 100 µM melatonin treatment, the soluble sugar content was significantly higher than that of S plants at 3, 6, and 12 d. Under the 500 µM melatonin treatment, the soluble sugar content was significantly higher than that of S plants at 6 d. Under the 200 µM melatonin treatment, the soluble sugar content of cotton leaves increased most significantly. The soluble sugar contents at 3, 6, 9, and 12 d corresponded to significant increases of 91.7%, 72.7%, 28.2%, and 38.4%, respectively, compared with S plants. The 200 µM melatonin treatment most obviously promoted the soluble sugar content of cotton leaves.

Compared with the CK treatment, the soluble protein content under the salt stress (S) treatment was significantly reduced, and the soluble protein content at 3, 6, 9, and 12 d decreased by 8.7%, 15.3%, 9.4%, and 16.4%, respectively (Fig. 5B). Under exogenous melatonin, the soluble protein content first increased and then decrease as melatonin concentration increased. Under the 50 µM melatonin treatment, the soluble protein content of cotton leaves was significantly higher than that of S plants only at 6 d; under the 100 and 500 µM melatonin treatments, the soluble protein content was significantly higher than that of S plants at 6, 9, and 12 d. The soluble protein content of cotton leaves increased most significantly under the 200 µM melatonin treatment, and the soluble protein contents at 3, 6, 9, and 12 d corresponded to significant increases of 7.1%, 17.5%, 9.1%, and 18.1%, respectively, compared with S plants, indicating that 200 µM melatonin effectively increased soluble protein content in cotton leaves.

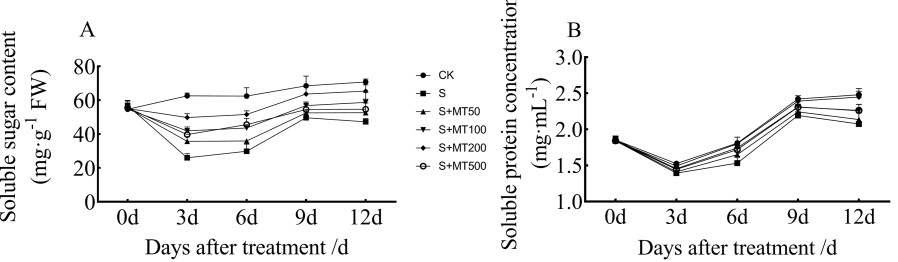

**Figure 5** **Effects of exogenous melatonin (MT) treatment on soluble sugar (A) and soluble protein (B) contents of cotton leaves under salt stress.** Control (CK) and salt-treated (S) plants were sprayed with distilled water, while S + MT50, S + MT100, S + MT200, and S + MT500 plants were sprayed with 50, 100, 200, and 500 μM MT, respectively.

## DISCUSSION

Salt damage is a major factor limiting the growth and yield formation of cotton seedlings (*Munns & Tester, 2008*). Salt stress can disrupt ion homeostasis in plants, cause ion poisoning, reduce the water and osmotic potentials of cells, and induce osmotic stress (*Abbasi et al., 2016*). When plants are subjected to salt stress, the most common major phenomenon is growth inhibition (*Towfique, Hafiz & Sen, 2013*). Melatonin is a highly conserved and physiologically active indoleamine plant hormone that can effectively alleviate damage caused by abiotic stress to plants (*Arnao & Hernández-Ruiz, 2015*). Several studies have reported that melatonin enhances abiotic stress resistance in naked oat (*Gao et al., 2019*), cucumber (*Zhang et al., 2014*), maize (*Li et al., 2019c*), and *Citrus aurantium* (*Kostopoulou et al., 2015*). In this study, melatonin treatment increased plant height and leaf area of cotton seedlings, which is consistent with the results of *Gao et al. (2019)*, who found that melatonin alleviates oxidative stress in naked oat under alkaline conditions.

Plant cells can maintain low ROS levels under normal growth conditions because the antioxidant enzyme system in the body achieves a balance between the elimination of ROS and the production of ROS. Under salt stress, plants accumulate high levels of ROS owing to excessive $Na^+$ accumulation, which destroys the ROS clearance systems of plants, leading to membrane lipid peroxidation, increased membrane permeability, and structural damage to membranes and membrane lipids (*Zhan et al., 2019*). The content of MDA, the final product of membrane lipid peroxidation, increased ROS accumulation under salt stress (*Hernandez et al., 2000*; *Møller, Jensen & Hansson, 2007*). In the present study, the $H_2O_2$, $O_2^-$, and MDA contents of cotton seedlings were remarkably increased by NaCl treatment. Over time, $H_2O_2$ and $O_2^-$ contents decreased, while MDA content increased, indicating that high levels of active oxygen were produced in cotton under salt stress, leading to membrane lipid peroxidation damage. The application of different concentrations of melatonin significantly reduced $H_2O_2$, $O_2^-$, and MDA contents (Fig. 2). This is consistent with findings reported for tea (*Li et al., 2019b*), naked oat (*Gao et al., 2019*), and rubber trees (*Yang et al., 2020*) under cold, salt, and drought stresses. Melatonin

directly scavenges $H_2O_2$ and enhances the activities of antioxidant enzymes to detoxify $H_2O_2$ indirectly; additionally, water balance is regulated under drought conditions by up-regulating the expression of melatonin synthesis genes (*Li et al., 2015*). This is probably because melatonin, as an electron donor, directly reacts with $H_2O_2$, hydroxyl radicals ($\cdot$ OH), and related molecules, and melatonin is itself oxidized to an indole cation free radical, which in turn reacts with intracellular $O_2$ and further oxidizes to stable *N*-acetyl-*N*-formyl-5-methoxykinamine (5-MAFK). Melatonin induced MAPK cascade related to $H_2O_2$ signaling pathway, promoted the expression of TFs WRKY genes related to salt stress (*Gao et al., 2019*) and alleviate membrane lipid peroxidation under stress (*Ye et al., 2016*; *Cui et al., 2017*), thereby increasing the tolerance of plants to salt. Thus, melatonin improves the ability of cotton seedlings to scavenge ROS and reduces membrane lipid peroxidation damage to cells. Moreover, the effect of melatonin is dependent upon its concentration. Among treatments, the 200 μM melatonin treatment exhibited the strongest ability to enhance ROS scavenging.

Plants can protect their cells from oxidative damage by removing excess ROS under adverse conditions through enzymatic and non-enzymatic antioxidant systems; the main antioxidant enzymes in plants are SOD, POD, CAT, and APX (*Li et al., 2019a*). Application of melatonin to tomato plants has been shown to minimize the negative impact of drought by regulating the antioxidant system, which reduces the abundance of toxic cells in plants, resulting in stronger, more drought-tolerant seedlings (*Liu et al., 2015a*; *Li et al., 2019b*). Melatonin could improve salt resistance by regulating the corresponding genes encoding antioxidant enzymes in cucumber under high salinity (*Zhang et al., 2014*). In the present study, the activities of SOD, POD, CAT, and APX in cotton seedlings increased significantly under salt stress and gradually decreased throughout treatment time. In the early stage of salt stress, CAT activity in cotton seedlings increased rapidly, but decreased gradually with time, indicating that salt stress inhibited CAT activity. After applying different concentrations of melatonin, the activity levels of SOD, POD, CAT, and APX were significantly increased (Fig. 3). These results were consistent with previous studies. Melatonin has strong antioxidant activity and increases antioxidant enzyme activity of cotton seeds and watermelon seedlings, thus reducing peroxidative damage (*Castañares & Bouzo, 2019*; *Chen et al., 2019*; *Xiao et al., 2019*). This may have been caused by exogenous melatonin upregulating the expression of genes related to antioxidant enzymes and reducing the degradation of biological macromolecules to improve the activity of antioxidant enzymes (*Zhang et al., 2014*). Exogenous melatonin significantly reduced malondialdehyde content and markedly increased the expression of antioxidant genes in melon (*Zhang, Yang & Chen, 2017*), wheat (*Sun et al., 2018*), hickory (*Wang et al., 2019*), and tea trees (*Li et al., 2019b*) under abiotic stress. In turn, this may thus have improved the ability of plants to remove ROS while alleviating salt stress in cotton seedlings caused by oxidative stress. Similarly, the effect of melatonin is dependent on its concentration. Among the treatments employed, the 200 μM melatonin treatment had the most significant effect on the promotion of antioxidant enzymes in cotton seedlings, which is consistent with the results of *Gong et al. (2017)*, who found that melatonin alleviates oxidative stress in *Malus hupehensis* under alkaline conditions.

Excessive ROS produced by plants under stress conditions are managed by the non-enzymatic systems of plants to protect their cells from oxidative damage (*Manchester et al., 2015*). AsA and GSH are the main reducing substances in plants, both of which can effectively remove excess ROS and $H_2O_2$ through the AsA-GSH cycle, thereby reducing damage to plants caused by adverse conditions (*Bybordi, 2012*). In tomato leaves under salt and alkali stress, AsA-GSH cycle components, including ascorbate peroxidase (APX), dehydroascorbate reductase (DHAR), and glutathione reductase (GR), were expressed at higher levels and played an important role in scavenging ROS (*Gong et al., 2013*). In the present study, the AsA and GSH contents of cotton seedling leaves increased significantly under salt stress and gradually decreased throughout treatment time. The decrease in AsA content indicated that some AsA was oxidized to dehydroascorbic acid (DHA) under salt stress. The decrease in GSH is most likely owing to GSH's involvement in ROS scavenging as a substrate for glutathione oxidase (GPX) or glutathione transferase (GST). After the application of different concentrations of melatonin, the AsA and GSH contents of each treatment increased significantly (Fig. 4). GSH exists in most organelles in cells and protects plants from oxidative damage caused by adversity stress (*Gill & Tuteja, 2010*). In *Citrus aurantium*, exogenous melatonin can effectively increase the content of reducing substances, such as phenols, AsA, and GSH, under salt stress and alleviate oxidative damage induced by salt stress (*Kostopoulou et al., 2015*). Exogenous melatonin enhances the tolerance of tomato and cucumber seedlings to stress by stimulating the AsA-GSH cycle enzyme activity and inducing the accumulation of AsA and GSH (*Liu et al., 2015b*; *Wang et al., 2016*; *Yin et al., 2019*). This may be caused by melatonin increasing monoascorbate reductase (MDHAR), DHAR, and GR activity levels and accelerating the regeneration of AsA and GSH, thereby ensuring relatively high concentrations of AsA and GSH in cotton under salt stress, and enhancing resistance to salt stress. Thus, melatonin alleviated oxidative stress caused by salt stress in cotton seedlings by increasing the content of reducing substances (i.e., AsA and GSH) and the activity of antioxidant enzymes. In addition, the relationship between melatonin and its effect depended on its concentration, and the 200 μM melatonin treatment most obviously promoted AsA and GSH, which is consistent with the findings of similar previous studies (*Shi et al., 2015*; *Zhao et al., 2016*).

Osmotic regulation is a key self-defense mechanism produced by plants under stress. Plants produce high levels of osmotic regulating substances (e.g., soluble sugars and proteins) to increase cytoplasmic solute concentrations, increase cell osmotic pressure to reduce water potential, and reduce cell water loss (*Verbruggen & Hermans, 2008*). Under salt stress, plants maintain necessary nutrients for their metabolic development by increasing osmotic regulators, thereby reducing damage caused by salt stress. A previous study has shown that adverse stress reduces the soluble sugar content of maize seedlings, while melatonin treatment increases soluble sugar content; soluble sugars participate in osmotic regulation of plants and alleviate the inhibitory effect of stress on maize growth (*Li et al., 2019c*). Most soluble proteins in plants are enzymes involved in various metabolisms (*Acostamotos et al., 2017*). In this study, the soluble sugar and soluble protein contents of cotton seedling leaves significantly decreased under salt stress. Over time, the contents of soluble sugars and proteins first increased and then decreased, possibly as a consequence of

salt stress. High levels of ROS destabilize sugars and proteins or affect their synthesis and metabolism pathways, causing the content of soluble sugars and proteins to decrease. After applying different concentrations of melatonin, the content of soluble sugar and protein in each treatment increased significantly, probably because melatonin promoted the synthesis of heat shock proteins in cotton seedlings and protected proteins from damage, indicating that melatonin participated in the osmotic adjustment of cotton seedlings (Fig. 5). These findings are similar to those presented by *Gao et al. (2019)*. Melatonin treatment can increase the soluble protein content of plants, alleviate damage to soybean and cotton seeds under drought and salt stress, and improve plant resistance to stress (*Gao et al., 2019*; *Chen et al., 2020*). Similarly, exogenous melatonin can alleviate osmotic stress caused by drought by increasing soluble sugar and soluble protein content of rapeseed, and it thus improves the growth of seedlings (*Li et al., 2018b*). The effect of melatonin on the osmotic adjustment of cotton seedlings was accomplished by increasing the content of soluble sugars and proteins, thus relieving the osmotic stress caused by high salt. Soluble sugar and protein contents under the 200 $\mu$M melatonin treatment increased the most, indicating that the effect of melatonin is related to its concentration, which is consistent with the results of *Li et al. (2018b)*, who showed that exogenous melatonin alleviates osmotic stress in rapeseed plants under drought conditions.

## CONCLUSIONS

Exogenous application of melatonin can improve the ROS scavenging capacity of cotton plants under salt stress by enhancing their antioxidant capacity. Additionally, melatonin treatment can alleviate osmotic stress by promoting the accumulation of osmoregulatory substances such as soluble sugars and proteins. Ultimately, exogenous melatonin could be used to facilitate development of cotton seedlings under salt stress, thereby alleviating salt stress in cotton seedlings. The 200 $\mu$M melatonin treatment is the most effective in promoting cotton seedling growth and salt tolerance in this experiment. The result provides a theoretical basis for melatonin to alleviate salt stress caused by unreasonable irrigation, fertilization and climate change. Further investigations need to be done to explore the molecular mechanism of exogenous melatonin regulating cotton seedlings under salt stress.

## ACKNOWLEDGEMENTS

The authors are grateful to the anonymous reviewers for their valuable comments and suggestions.

### Funding

This work was supported by the National Natural Science Foundation of China (No. 31871569, Liantao Liu), the Independent Research Project in State Key Laboratory of North China Crop Improvement and Regulation (No.NCCIR2020ZZ-18, Zhiying Bai),

and the Key Research and Development Project of Hebei Province (No. 20326409D, Zhiying Bai). The funders had no role in study design, data collection and analysis, decision to publish, or preparation of the manuscript.

### Grant Disclosures
The following grant information was disclosed by the authors:
National Natural Science Foundation of China: 31871569.
Independent Research Project in State Key Laboratory of North China Crop Improvement and Regulation:  NCCIR2020ZZ-18.
Key Research and Development Project of Hebei Province:  20326409D.

### Competing Interests
The authors declare there are no competing interests.

### Author Contributions
- Dan Jiang, Bin Lu and Liantao Liu conceived and designed the experiments, performed the experiments, analyzed the data, prepared figures and/or tables, and approved the final draft.
- Wenjing Duan, Li Chen and Jin Li analyzed the data, prepared figures and/or tables, and approved the final draft.
- Ke Zhang, Hongchun Sun, Yongjiang Zhang and Hezhong Dong conceived and designed the experiments, authored or reviewed drafts of the paper, and approved the final draft.
- Cundong Li and Zhiying Bai conceived and designed the experiments, performed the experiments, authored or reviewed drafts of the paper, and approved the final draft.

### Data Availability
The raw measurements are available in the Supplementary Files.

### Supplemental Information
Supplemental information for this article can be found online at http://dx.doi.org/10.7717/peerj.10486#supplemental-information.

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
