# Peer review of "Exogenous melatonin improves salt stress adaptation of cotton seedlings by regulating active oxygen metabolism"

_PeerJ, doi:10.7717/peerj.10486_

## Round 0.1 · original submission · Major Revisions

Although the reviewers see value in your data, you need to make the scientific contribution of your study more clear, for example by defining your question.

·

Basic reporting

no comment

Experimental design

no comment

Validity of the findings

no comment

Additional comments

The manuscript entitled “Exogenous melatonin improves salt stress adaptation of cotton seedlings by regulating active oxygen metabolism” has been submitted to Peer J. The study has been conducted well but some concerns make me hesitant. There are some points that I think might be helpful.
The novelty and hypothesis of the study have not been defined. There are some experiments that showed the effects of melatonin as a ROS scavenger, so, please mention what can distinguish your work.
The introduction should be re-written because it emphasizes the effects of salinity and antioxidant enzyme activities which is completely clear for scientists. The most important part of your work is the application of melatonin, not salinity. What is the melatonin, mechanism of action, biosynthesis pathway and …
Why did you select 150 mM?
I think you should have had control+ 50, 100, 200 and 500 melatonin as treatments, which could have shown the effects of melatonin under normal conditions. Why did you delete these treatments?
Line 132-133 “When the cotton seedlings reached the three-true-leaf stage, the following treatments were imposed” how many days after transplanting?
The statistical design should be clarified. One-way ANOVA is not correct. Salinity and melatonin are two factors.
How did you compare means? LSD, Tukey?
Use one decimal for percentage. 37.2% instead of 37.24%
How many times did you spray MT?
The quality of figures is low, please replace them with high-quality versions.
Your work is a physiological study, so, the discussion should be explained the pathways and mechanisms of actions, therefore, please improve this part.
Line 391: “This study showed that …” please revise the sentence.
Line 419-421: “was oxidized to oxidized ascorbic acid (DHA)…” should be revised.
Line 461: “In conclusion” should be removed.
The conclusion should be re-written. No recommendation and suggestion. Please don’t repeat your results.

Reviewer 2 ·

Basic reporting

The present manuscript explains the effect of exogenous melatonin in cotton under salt stress by measuring different parameters. The study is well-structured but some changes are needed.

First, the English language should be improved to ensure that an international audience can clearly understand your text.

Experimental design

The experimental design is correct.

Line 130-131: Use micromolar as concentration unit in micronutrients.
Line 160-164: Follow the same style indicating the assay kit reference as for soluble sugar and soluble proteins.
Line 153, 161, 167, 169: change to "by using ...".

Validity of the findings

The results part could be improved on the result presentation as there are many data (so many percentages, which can be confused). A possibility is to add a PCA analysis to show how diferent parameters are affected by melatonin addition as general vision of melatonin addition under salt stress.

Discussion part need to be re-structure as the authors presented results as in the Result part and then, they explained effects of melatonin or salt stress in other plants. It could be better if they explain both together. Justification of results need to be improved.
Also, some sentences need a reference (ex. line 441).

Line 177-178: This text has no sense. Re-write.
Line 197-199: Add reference.
Line 201: Use past simple.
Line 240-242: Add reference.
Line 316-318: Add reference.

Additional comments

The article need to change some aspects before acceptacion.

Reviewer 3 ·

Basic reporting

The role of secondary metabolites are well known in improving plant resistance to abiotic stresses. The functional role of melatonin in stress regulation is less known and seldom studied. In this perspective, the present study is important. The contents of the paper are quite well structured and nicely presented. Results are no doubt important to understand the stress responses and to evaluate the role of melatonin in stress regulation. However, the conclusion is drawn on the basis of basic experimental findings and such approach have been reported in plants under stress. The work is nice, but lacks novelty and individuality.

Experimental design

We designed but lacks novel experimental approach.

Validity of the findings

The biochemical datasets depicts the role of melatonin in controlling salinity stress in cotton. However, only biochemical datasets cannot be decisive in concluding that melatonic controls the redox regulation. Other more precise approaches could have been adopted to come up with a concrete conclusion.

Additional comments

1. The paper lack novelty from experimental approach point of view.
2. The authors may use other approaches such as antioxidant gene expression profiles, redox proteomics and metabolomics approach supported by biochemical analysis.

---

## Round 0.2 · Minor Revisions

Your manuscript has improved substantially, and the message is clear. However, you should rewrite your final conclusion: 'The 200 μM melatonin treatment is most effective in promoting cotton seedling growth and salt tolerance.' You could claim, e.g., that treatment with 200 μM WAS the most effective in your experiments.

You also could speculate about the usability of your protocols for the cultivation of other plants.

Further, you could emphasize the importance of your study for future agriculture with increased salinity of soils due to irrigation (climate change).

·

Basic reporting

The study conducted well, and the manuscript has been written in a good shape, especially after revision. English is at a standard level for publication.

Experimental design

Materials and methods have been significantly improved.

Validity of the findings

My biggest concern about this study is the novelty; there are several papers that evaluated the effects of melatonin as a salt alleviator.

The conclusion needs more improvements. You should present suggestions and recommendations to researchers or drow a clear picture of future studies.

Additional comments

After revision, the manuscript has been improved significantly. The goals and hypotheses have been mentioned clearly. All comments have been addressed completely.

---

## Round 0.3 · Minor Revisions

Thank you for revising your manuscript again and for contributing to the important topic of improving the cultivation of plants under high-salt conditions. The section editor has some additional comments:

1) "The experimental design is unclear. What was the blocking design for the various treatments? i.e. were all of the plants subject to the "S+MT50" treatment grouped together or were they randomized with
respect to the other treatments? "

2) "+++ line 153 "six cotton seedlings with uniform growth". Why did they pre-select for uniform growth if you are trying to measure leaf size and plant height? This introduces biases into the experiment."

Could you please answer these two points prior to acceptance of the paper?

---

## Round 0.4 · Minor Revisions

Dear authors, thank you for replying to the additional comments in your rebuttal letter. However, the additional information also should be included in your manuscript.

---

## Round 0.5 · accepted · Accept

Thank you very much for including the additional comments into your manuscript.